# The Impact of Flow on University EFL Learners’ Psychological Capital: Insights from Positive Psychology

**DOI:** 10.3390/bs15121703

**Published:** 2025-12-08

**Authors:** Fan Jia, Xihong Wang, Chunjie Ding, Shujun Wang, Xiaorong Wang, Yanhui Mao

**Affiliations:** 1School of Foreign Language, Southwest Jiaotong University, Chengdu 610031, China; jiafan@swjtu.edu.cn; 2Institute of Applied Psychology, Psychological Research and Counseling Center, Southwest Jiaotong University, Chengdu 610031, China; xihongwang@my.swjtu.edu.cn (X.W.); dingcj@my.swjtu.edu.cn (C.D.); shujun.wang@my.swjtu.edu.cn (S.W.)

**Keywords:** flow, foreign language classroom anxiety, academic efficacy, psychological capital

## Abstract

Many studies have shown that flow, psychological capital (PsyCap), anxiety, and academic efficacy play significant roles in EFL learning, yet little attention has been paid to how these positive and negative states jointly shape learners’ PsyCap. Grounded in the broaden-and-build theory, this study investigated how flow, a state of deep engagement and enjoyment in learning, affected EFL learners’ PsyCap. A total of 1611 EFL learners at the CEFR B1–B2 levels from six universities in China participated in the study. Data were collected using validated questionnaires developed for this study that measured flow, foreign language classroom anxiety (FLCA), academic efficacy, and PsyCap, and analyzed using structural equation modeling (SEM) in AMOS. The results revealed that flow had a significant direct positive effect on PsyCap (*β* = 0.648, *p* < 0.001). Academic efficacy significantly mediated this relationship (*β* = 0.059, *p* < 0.001), and a significant chain-mediated path was observed through FLCA and academic efficacy (*β* = 0.023, *p* < 0.001). The total effect of flow on PsyCap was 0.729 (*p* < 0.001). These findings provide new insights into educational practices that can effectively enhance EFL learners’ PsyCap and academic achievement by facilitating flow and reducing anxiety.

## 1. Introduction

In the traditional foreign language classroom, teachers focus on imparting knowledge and skills while paying little attention to learners’ psychological state and intrinsic motivation, resulting in anxiety and burnout in the learning process, which in turn may contribute to classroom avoidance and decreased academic performance ([39]; [68]). In recent years, the rise of positive psychology has provided new insights for second language acquisition (SLA) research. While focusing on improving language performance, it also emphasizes cultivating positive psychological qualities to enhance learning effectiveness and subjective well-being, leading to a “positive shift” in EFL teaching and learning ([20]).

Flow, as an essential construct of positive psychology, has a positive impact on the cognitive development and emotional experience of EFL learners, and its focus on the individual’s internal experience and motivational stimulation provides a new perspective on language learning ([7]). When EFL learners enter a state of flow, their intrinsic motivation and learning engagement increase substantially, helping them achieve a deeper level of language acquisition ([23]; [80]). Moreover, flow has been associated with reduced anxiety and stress, and with improved enjoyment and academic performance ([27]; [65]). Thus, flow contributes meaningfully to both improved learning performance and more positive emotional experiences in EFL learning classrooms.

While flow highlights the importance of optimal experience and intrinsic motivation in language learning, another crucial factor contributing to learners’ emotional well-being and academic success is psychological capital (PsyCap). It is a core positive psychological resource that enhances individual resilience, well-being, and growth. PscyCap significantly influences academic achievement and the development of effective coping skills ([34]). In EFL learning, it positively affects emotional experiences. By building PsyCap, learners can reduce anxiety and boredom in the classroom, thereby increasing their motivation to learn ([30]). Additionally, higher levels of PsyCap improve adaptability and coping mechanisms in demanding learning environments, enhancing overall classroom performance ([45]). Overall, PsyCap supports EFL learners by fostering resilience and intrinsic motivation, thereby enhancing both their academic performance and classroom experience.

Research on self-efficacy has been a productive field in positive psychology and SLA. However, relatively few studies have examined how language learners develop academic efficacy ([6]). Given this gap, academic efficacy has garnered significant attention as a crucial determinant of students’ success in language learning, particularly in research on academic burnout, in which academic efficacy reflects students’ perceptions of their abilities in the academic field and their beliefs in their capacity to cope with academic challenges effectively ([63]). Defined as a student’s belief in their ability to achieve foreign language learning goals, academic efficacy is rooted in Bandura’s self-efficacy theory ([64]), which posits that an individual’s beliefs about their capabilities influence their motivation, persistence, and ultimately, their performance in language acquisition tasks ([6]). Research has consistently demonstrated that academic efficacy plays a vital role in shaping students’ approaches to foreign language learning and their ability to overcome language-related challenges ([2]). For instance, studies have shown that higher levels of academic efficacy are associated with greater proficiency in the target language, increased engagement with foreign-language learning materials, and better stress management in language-learning settings ([54]). Moreover, academic efficacy has been found to influence various aspects of the foreign language learning experience, including students’ selection of language-learning tasks, their effort expenditure, and their resilience in the face of setbacks ([75]). Understanding and fostering academic efficacy is thus essential for EFL educators aiming to support students in achieving their full potential in language learning.

The phenomenon of foreign language classroom anxiety (FLCA) has garnered significant attention due to its pervasive impact on learners’ experiences and outcomes. FLCA refers to the tension and worry learners experience in foreign language learning situations, especially during speaking and listening activities ([38]). Research indicated that a substantial proportion of students, approximately one-third of Chinese college students, report experiencing anxiety in English classes ([5]). However, recent research has highlighted the potential of PsyCap and the concept of flow to counteract the negative effects of FLCA. PsyCap has been shown to enhance learners’ ability to regulate anxiety and maintain a positive attitude towards learning ([51]). Understanding the interplay between FLCA, PsyCap, and flow is essential for educators aiming to foster a more positive and productive learning environment for EFL learners. By developing strategies to enhance PsyCap and promote flow, educators can help students better manage their anxiety and achieve their full potential in language learning ([51]). This study highlights the importance of flow, PsyCap, and FLCA during EFL learners’ formation of academic efficacy. Implications are provided for reducing EFL learners’ physiological stress and sustaining their academic efficacy for improving their PsyCap.

To summarize, many studies have demonstrated that flow, PsyCap, anxiety, and academic efficacy are beneficial to L2 learning. However, little attention has been paid to how these positive and negative states jointly shape learners’ PsyCap development. In other words, empirical studies exploring their integrated psychological mechanism remain limited. Prior studies have mainly examined dyadic relationships (e.g., flow–motivation, anxiety–performance), but few have tested the chain mediation among these positive and negative psychological factors in EFL contexts. Examining the chain mediation allows us to uncover how flow exerts its influence through emotional and motivational processes, thereby clarifying the underlying mechanism through which short-term experiences translate into long-term psychological resources such as PsyCap. Based on this, this study integrates academic efficacy and FLCA into a single model to clarify how flow can foster enduring PsyCap. This framework addresses a gap in understanding how momentary experiences shape long-term psychological growth in EFL learning within Chinese cultural contexts. This research perspective not only helps to enrich the theoretical foundation in the field of flow and PsyCap but also provides new insights into educational practices that can effectively enhance EFL learners’ PsyCap and academic achievement by facilitating flow and reducing anxiety.

## 2. Conceptual and Theoretical Framework, Literature Review, and Hypotheses

### 2.1. Theoretical Foundation

This study is based on the broaden-and-build theory (BBT), which argues that positive emotional experiences expand an individual’s thinking and actions. Positive emotions can initiate cycles of beneficial emotions and behaviors through dynamic psychological processes, significantly influencing long-term emotional health and well-being ([26]). This contributes to the development of enduring personal resources, including physical, intellectual, social, and psychological assets ([24]).

Within this framework, flow functions as a transient positive state that broadens learners’ attention and engagement, enhancing their sense of mastery and academic efficacy ([33]). At the same time, these positive experiences help attenuate foreign language classroom anxiety, a debilitating negative emotion that restricts learners’ cognitive resources ([68]). Furthermore, hope—as defined in the BBT—is a positive emotion that arises in challenging situations and motivates individuals to pursue more favorable outcomes ([25]). When repeatedly experienced through flow, this momentary hope can gradually evolve into the trait-like hope component of psychological capital ([47]).

Therefore, the BBT offers an integrative explanation for how positive emotions (flow, academic efficacy) and negative emotions (anxiety) interact in EFL learning to influence learners’ PsyCap development. This theoretical framework not only reveals these dynamic relationships but also guides the optimization of EFL instruction by creating contexts that foster flow and support the development of psychological resources.

### 2.2. Constructs and Related Studies

#### 2.2.1. Flow

Flow is a mental state in which a person is fully immersed in an activity, experiencing energized focus, full involvement, and enjoyment ([15]). In EFL research, flow has been viewed as a relevant variable for predicting students’ interest, teachers’ teaching effectiveness, and the creation of an optimal learning state ([37]).

Empirical studies have shown that flow can improve learning efficiency and engagement by driving intrinsic motivation and alleviating external pressure ([28]). Furthermore, creating activities that promote flow can help improve EFL learners’ input skills (i.e., reading), enhance their sense of efficacy ([65]), help them clarify their learning goals for output skills (i.e., writing), and improve their resistance to interruptions ([46]). In addition, flow promotes intrinsic motivation and sustained cognitive involvement, creating learning conditions that support deeper processing and greater academic persistence. Evidence from weekly diary studies and scale-validation work further indicates that flow strengthens learners’ self-beliefs and supports optimal academic functioning ([36]; [37]). However, although studies have linked flow to improved motivation and emotional well-being, much less is known about the emotional mechanisms through which flow contributes to long-term psychological development. This gap highlights the need to examine flow within a more integrative emotional and motivational framework.

#### 2.2.2. Psychological Capital (PsyCap)

As one of the key psychological resources supporting optimal learning experiences, positive psychological resources have received growing attention in EFL research. Among these resources, psychological capital (PsyCap) is regarded as a core component. PsyCap, proposed by [47] ([47]), refers to a set of positive psychological states—self-efficacy, hope, optimism, and resilience—that individuals exhibit during their growth and development, and it includes a range of psychological resources that can be developed and nurtured. In educational settings, abundant PsyCap has been associated with greater resilience, enhanced motivation, and improved emotional well-being, making it a meaningful construct for understanding learners’ long-term psychological functioning.

A growing body of empirical work has shown that PsyCap benefits EFL learners in multiple ways. Higher levels of PsyCap predict greater behavioral, emotional, and cognitive engagement ([19]; [41]) and are linked to higher academic enjoyment and lower academic boredom ([73]). PsyCap also enhances learners’ adaptability and coping strategies when dealing with stressors in language learning ([45]). Moreover, PsyCap contributes to learners’ persistence, motivation, and willingness to engage with challenging tasks ([40]). However, little is known about how PsyCap develops through emotional processes, particularly when positive states (e.g., flow) and negative emotions (e.g., anxiety) are considered together. This limitation underscores the need for a model that integrates emotional and cognitive variables to clarify the developmental mechanisms of PsyCap in EFL contexts.

#### 2.2.3. Academic Efficacy

In the context of English as a Foreign Language (EFL) learning, “academic” is often used to describe language learning that occurs within educational institutions, as opposed to naturalistic or informal language acquisition. It includes classroom-based instruction, teacher-assigned tasks, assessments, and students’ perceptions of their learning experiences in these settings ([6]; [63]). For example, academic efficacy refers to students’ beliefs in their ability to succeed in academic-specific tasks, such as completing assignments, passing exams, or mastering course content. These beliefs are shaped by academic feedback, prior performance, and instructional support, and they directly influence students’ motivation, persistence, and emotional responses in educational environments ([55]; [79]).

Academic efficacy, rooted in Bandura’s self-efficacy theory ([64]), is not merely a general feeling of confidence but is grounded in perceived skills, knowledge, and past experiences within academic contexts ([79]). According to Bandura, efficacy beliefs are shaped by reciprocal influences among personal, behavioral, and social/environmental factors. Several antecedents influence the development of academic efficacy. For instance, past academic performance plays a significant role, as consistent success in previous courses or assignments reinforces students’ belief in their academic capabilities ([55]). Academic efficacy is positively related to language learning achievement. Students with higher academic efficacy are more likely to put in greater effort, persist in the face of challenges, and engage more deeply with language learning tasks ([4]). Additionally, they are more likely to be intrinsically motivated to learn and actively participate in classroom activities ([54]).

Despite growing interest in positive-psychology constructs, two critical gaps remain. First, no study has tested whether domain-specific academic efficacy functions as a mechanism through which flow experiences translate into higher-order psychological capital in EFL settings: a. Although [57] ([57]) linked flow to self-efficacy and [47] ([47]) located efficacy at the core of PsyCap, their integrated causal chain has yet to be examined. Second, while FLCA is known to impede language learning ([35]; [38]) and covary with flow ([50]), its sequential role—as a barrier that must be reduced by flow before academic efficacy and PsyCap can accrue—has never been modeled. The present study addresses both lacunae by simultaneously evaluating (1) a direct resource-building path (flow → academic efficacy → PsyCap) and (2) a barrier-removal path (flow → FLCA → academic efficacy → PsyCap), thereby offering the first comprehensive test of how optimal experience fosters sustainable psychological resources in SLA.

#### 2.2.4. Foreign Language Classroom Anxiety (FLCA)

FLCA is a complex phenomenon that significantly impacts language learning, rooted in [35]’s ([35]) seminal work. Anxiety is a situation-specific emotional reaction characterized by apprehension, tension, and worry over perceived threats to one’s self-worth or performance ([35]). Within formal educational settings, anxiety typically surfaces when learners anticipate negative evaluation, fear failure, or doubt their ability to meet academic demands ([77]). In the foreign-language classroom, these feelings are amplified by the need to communicate in an unfamiliar linguistic code, giving rise to what is known as foreign language classroom anxiety (FLCA). FLCA refers to the tension and worry learners experience in foreign language learning situations, particularly during speaking and listening activities ([38]). This anxiety is not merely a fleeting emotional state but a complex construct that encompasses self-perceptions, beliefs, feelings, and behaviors related to the unique challenges of language learning ([31]).

It is argued that the consequences of FLCA are far-reaching. Empirical studies have shown that high levels of anxiety can negatively impact language learning achievement by reducing learners’ willingness to communicate and participate in class ([76]). For example, anxious students may avoid speaking activities, leading to fewer opportunities for practice and feedback ([74]). In summary, FLCA has significant implications for language learning outcomes and classroom engagement. By understanding these dynamics, educators can implement strategies to create a more supportive learning environment and help reduce learners’ anxiety ([21]).

Despite extensive research identifying Foreign Language Classroom Anxiety (FLCA) as a complex, context-specific construct ([31]; [35]; [38]) with detrimental effects on language learning engagement and achievement ([76]; [74]), critical gaps remain: (a) unclear causal direction and interactive mechanisms between FLCA and flow in EFL contexts, despite flow’s potential to mitigate anxiety ([21]); (b) no empirical testing of FLCA’s sequential mediating role alongside academic efficacy in the “flow→PsyCap” pathway; and (c) insufficient exploration of contextualized variations in FLCA’s impact (e.g., digital vs. in-person settings; [77]) and the mechanisms through which it constrains key psychological resources like academic efficacy and PsyCap.

### 2.3. The Relationship Between Flow and PsyCap

Prior studies have explored how flow relates to different components of psychological capital. For example, [57] ([57]) investigated the relationship between flow and self-efficacy, and found that EFL learners’ classroom flow can lead to positive perceptions of their ability to cope with academic challenges and enhance their self-efficacy. Similarly, [9] ([9]) focused on the connections between flow, self-efficacy, and optimism, finding that elementary school teachers’ flow could positively predict their teaching efficacy and academic optimism. In addition, [48] ([48]) investigated the relationship between flow and resilience and found that college students’ flow could help enhance their psychological resilience and sense of well-being.

It can be concluded that many studies have highlighted the significant positive correlations between flow and the individual components of PsyCap. However, few have examined the link between flow and PsyCap as a whole construct, which is essential for learners’ sustained engagement and success in EFL. Given the complexity of this relationship, it is also essential to consider the cross-cultural cognitive challenges specific to EFL contexts, which may influence the pattern of correlation between the two. Based on this, the present study proposes Hypothesis 1: Learners’ flow during EFL learning can significantly positively predict their PsyCap level.

### 2.4. Path Mediation of Academic Efficacy Between Flow and PsyCap

Academic efficacy refers to perceived capabilities to learn or perform actions at designated levels in academic settings. When students have high academic efficacy, they are more likely to feel confident in their academic abilities, which directly contributes to overall PsyCap. This confidence can also enhance their motivation to engage in learning activities and pursue academic goals, a key aspect of PsyCap ([44]). Students with high academic efficacy are more likely to persist in the face of challenges and setbacks. This resilience is a core component of PsyCap and helps students maintain a positive outlook and continue to strive for success even when they encounter difficulties ([1]). Research on PsyCap suggests that individuals with high levels of efficacy are also likely to have high levels of the other PsyCap dimensions, namely hope, optimism, and resilience ([61]).

Flow is related to student well-being and academic success ([13]). When students enter a flow state during learning, they tend to feel a stronger sense of efficacy ([10]). Although academic efficacy permeates many aspects of students’ academic lives, experiencing flow can temporarily heighten their confidence and reduce self-consciousness during learning ([14]). Flow encourages students to adopt more effective learning strategies, such as active learning and metacognitive monitoring. Optimizing these strategies helps students complete learning tasks more efficiently, thereby improving academic efficacy ([10]). As an essential component of flow, autonomy or the freedom of individuals to arrange activities has been repeatedly found to increase positive affect and motivation ([50]). Experiencing flow can temporarily enhance students’ confidence and reduce self-consciousness during academic tasks, thereby reinforcing their academic efficacy ([14]). Therefore, this study proposes Hypothesis 2: Academic efficacy significantly mediates the relationship between flow and PsyCap; that is, flow significantly improves academic efficacy, which in turn significantly enhances an individual’s PsyCap.

### 2.5. The Role of FLCA and Academic Efficacy in the Relationship Between Flow and PsyCap

FLCA is an anxious experience specific to EFL learners, which may affect students’ motivation, self-efficacy, and academic performance ([78]). Previous research has shown that positive flow, which lies between anxiety and boredom, is a desirable and challenging learning state that stimulates students’ interest and intrinsic motivation ([17]). According to the affective filter hypothesis, anxiety hinders language input during foreign language learning, thus affecting learning effectiveness ([22]). FLCA, as a significant psychological factor affecting students’ academic efficacy, has garnered widespread attention from educators and researchers ([66]). In the process of learning English, students often experience significant anxiety due to language expression barriers and fear of failure ([56]). Experiencing a positive flow, EFL learners feel more active, focused, happy, and satisfied, which can reduce anxiety and enhance learning ([16]). Flow also enhances students’ efficacy and makes them more confident in facing difficulties in learning, further reducing anxiety levels ([78]).

In addition, FLCA may weaken students’ efficacy and motivation to learn ([66]), which in turn may further reduce students’ PsyCap, and learners with high PsyCap tend to have higher self-efficacy, stronger hope, greater resilience, and more optimistic attitudes ([53]), and these traits help reduce FLCA. Students who experience flow more frequently in daily life are more likely to have higher self-esteem and lower anxiety levels. Individuals who experience flow more often across domains also tend to have fewer depressive symptoms and less emotional exhaustion ([52]). Therefore, we propose Hypothesis 3: FLCA and academic efficacy sequentially and significantly mediate the relationship between flow and PsyCap. Specifically, flow first significantly alleviates FLCA; this anxiety reduction subsequently significantly strengthens academic efficacy, which in turn significantly elevates PsyCap.

Based on the three research hypotheses proposed above, this study was constructed to address the following research questions by testing the following hypothetical model (see Figure 1).

## 3. Methods

### 3.1. Participants and Procedure

A total of 1669 undergraduate EFL learners from six public universities across eastern, central, and southwestern China participated in this study. Both English and non-English majors were included, all of whom were enrolled in compulsory English courses. The participants were recruited through convenience sampling ([43]), and instructors distributed the survey QR code during class time. To ensure data quality, responses that failed to answer trap questions or showed identical answers were excluded from the analysis. A final valid sample of 1611 participants remained. The age range of the participants was between 16 and 28 years old (*M* = 19.166, *SD* = 1.220). In terms of gender distribution within the sample, there were 757 males, accounting for 46.99%, and 854 females, accounting for 53.01%.

Before data collection, ethical approval was obtained from the ethical review boards of all participating universities. Participants were provided with detailed information about the study’s purpose, procedures, data confidentiality, and their rights. It was explicitly stated that participation was entirely voluntary, that they could withdraw at any time without penalty, and that their decision would not affect their course grades or relationship with the instructors. Furthermore, we assured participants that there were no right or wrong answers to the questionnaire items. A QR code linked to the questionnaire was shared with the participants, who were invited to scan the code and complete the survey voluntarily.

### 3.2. Instruments

#### 3.2.1. Flow Scale (State)

The Flow Scale (state) adapted from [71]’s ([71]) 8-item scale based on [14]’s ([14]) flow theory was employed to measure participants’ flow. Participants were instructed to reflect on their recent experiences in English learning and rate their agreement with each statement on a 7-point Likert scale, ranging from 1 (strongly disagree) to 7 (strongly agree). To ensure linguistic and conceptual equivalence, we adopted the Chinese translation of the scale used in [49] ([49]), which has been validated across American, Chinese, and Spanish samples (α = 0.77, 0.88, and 0.82, respectively; [49]; [11]; [36]). Based on the results of the factor analysis, item 2 (loading = 0.25), item 4 (loading = 0.24), and item 7 (loading = 0.21) were excluded from the analysis, as their factor loadings were below the recommended threshold of 0.40 ([32]). According to George and Mallery’s criteria (2003), the scale demonstrated excellent reliability, with a Cronbach’s alpha of 0.908 in the current sample. In addition, the CFA results indicated an acceptable model fit (*χ*^2^/*df* = 1.307; GFI = 0.999; CFI = 1.000; NFI = 0.999; TLI = 0.999; SRMR = 0.034; RMSEA = 0.014) ([62]; [72]).

#### 3.2.2. Foreign Language Classroom Anxiety Scale (FLCAS)

Participants’ FLCA was measured using the Foreign Language Classroom Anxiety Scale (FLCAS) developed by [35] ([35]). The scale consists of 33 items designed to reflect participants’ classroom experiences. Responses were recorded on a 5-point Likert scale ranging from 1 (strongly disagree) to 5 (strongly agree). Based on George and Mallery’s criteria (2003), the scale demonstrated excellent reliability, with a Cronbach’s alpha of 0.934 in the current sample. 

#### 3.2.3. Academic Efficacy Scale

The Academic Efficacy Scale was utilized to assess students’ perceived ability to manage academic challenges and their satisfaction with their accomplishments ([63]). The scale comprises six items. Participants respond to the items using a seven-point Likert scale, ranging from 0 (never) to 6 (every day), with higher scores indicating greater professional efficacy. In the current study, the Cronbach’s α coefficient for this subscale was 0.906, indicating excellent internal consistency and reliability ([29]).

#### 3.2.4. Psychological Capital Questionnaire

To measure PsyCap in this study, we utilized the Psychological Capital Questionnaire (PCQ) developed by [47] ([47]). The scale is a 24-item Likert-type scale (1 = strongly disagree, 6 = strongly agree) that evaluates four key dimensions: self-efficacy (items 1–6), hope (items 7–12), resilience (items 13–18), and optimism (items 19–24). This instrument has been extensively validated and is widely recognized for its reliability and validity across diverse samples ([12]; [59]; [67]; [3]). In our sample, the PCQ’s internal consistency was confirmed with a Cronbach’s alpha of 0.944, indicating excellent reliability ([29]).

### 3.3. Data Analysis

Firstly, we tested the construct reliability by Cronbach’s alpha. Then, we used Harman’s single-factor test to examine common-method bias. Pearson bivariate correlation analyses between flow, FLCA, academic efficacy, and PsyCap were also computed. All these tests were performed within SPSS (29.0).

Next, we performed confirmatory factor analysis (CFA) within AMOS (version 25) to test our measurement model, considering the latent variables with their corresponding observed indicators: Flow, FLCA, academic efficacy, and PsyCap. CFA allowed us to assess the reliability and validity of these constructs before testing the structural relationships ([32]).

Finally, a structural equation modeling using AMOS (version 25) was employed to conduct a chained mediation effect test, as it allows for the simultaneous estimation of multiple latent variables and indirect effects while controlling for measurement error ([42]). In this model, flow was the independent variable, FLCA and academic efficacy the mediating variables, and PsyCap the dependent variable. Then, the bootstrap method was used to test the chain mediation effect by resampling 5000 times to calculate the 95% confidence intervals.

## 4. Results

### 4.1. Common Method Bias

The results of the single-factor test showed that 9 factors had eigenvalues greater than 1, and the variance explained by the first unrotated factor was 30.969%, which is below the critical value of 40%. The results indicate that there is no serious common method bias in this study ([58]).

### 4.2. Descriptive Statistics and Correlations Among Study Variables

Table 1 indicates that flow was significantly negatively correlated with FLCA (*r* = −0.456, *p* < 0.001). PsyCap was significantly positively correlated with flow (*r* = 0.672, *p* < 0.001) and academic efficacy (*r* = 0.456, *p* < 0.001). Academic efficacy was significantly negatively correlated with FLCA (*r* = −0.370, *p* < 0.001).

### 4.3. Test of Our Measurement Model

CFA was performed to test the discriminant validity of the four main latent variables (flow, FLCA, academic efficacy, and PsyCap) using the 18 indicators comprising the measurement model (see Figure 2). The CFA results were all within the recommended thresholds ([62]; [72]), indicating an acceptable fit to our data: *χ*^2^/*df* = 4.550; GFI = 0.964; CFI = 0.982; NFI = 0.978; TLI = 0.976; SRMR = 0.036; RMSEA = 0.047. All the standardized factor loadings for the indicators of constructs were above the cut-off value (0.40) and significant (*p* < 0.001), suggesting that the constructs had good convergent validity. Specifically, the loadings ranged from 0.79 to 0.85 for flow, from 0.65 to 0.89 for FLCA, from 0.83 to 0.95 for academic efficacy, and from 0.77 to 0.96 for PsyCap (see Figure 2 for detailed values). To address the potential influence of using different response scales across measures (the FKCAS was rated on a 5-point scale, whereas the other instruments used 7-point scales), we conducted an additional robustness check. Specifically, we re-estimated the mediation model using all items as observed indicators. The structural paths remained highly consistent with those from the latent-variable model. Key paths such as Flow → FLCA (*β* = –0.231, *p* < 0.001), Flow → academic efficacy (*β* = −0.373, *p* < 0.001), FLCA → academic efficacy (*β* = 0.466, *p* < 0.001), Flow → PsyCap (*β* = 0.352, *p* < 0.001), and academic efficacy → PsyCap (*β* = −0.097, *p* < 0.001) all remained statistically significant. This convergence suggests that the discrepancy in response formats did not materially bias the substantive conclusions. Taken together, these indices demonstrated that the latent variables were indeed distinct constructs represented by their respective observed indicators; therefore, the latent variables can be further analyzed for path analyses.

### 4.4. Test of the Mediation Model via Structural Equation Modeling

Firstly, it was found that the mediation model had an acceptable fit index ([62]; [72]): *χ*^2^/*df* = 4.472; GFI = 0.963; CFI = 0.982; NFI = 0.977; TLI = 0.977; SRMR = 0.037; RMSEA = 0.046. Secondly, the mediation effect analysis results are presented in Table 2.

The direct effect of flow on PsyCap is statistically significant, with a 95% confidence interval that does not include zero (95% CI = [0.569, 0.722]). At the same time, the partial mediation of FLCA and academic efficacy play between flow and PsyCap were confirmed. The total indirect effect was 0.081, which is significant and accounts for approximately 11% of the total effect. Specifically, there are two paths that reach a significant level: the simple mediating effect through flow → academic efficacy → PsyCap (95% CI = [0.036, 0.084]), and the chain mediating effect through flow → FLCA → academic efficacy → PsyCap (95% CI = [0.014, 0.034]). The path diagram and standardized path coefficients (β) are shown in Figure 3.

## 5. Discussion

### 5.1. The Direct Predictive Effect of Flow on PsyCap

The results showed that EFL learners’ flow significantly and positively predicted their PsyCap, further supporting the BBT’s view that positive emotions broaden individuals’ momentary thought and action repertoires and build enduring psychological resources ([24]). Flow exemplifies a positive state in which learners’ flow helps regulate emotions and enhance engagement, and, over time, these positive emotional and cognitive expansions contribute to building more stable psychological resources, reflected in the higher levels of PsyCap observed in this study.

This result is consistent with previous studies, as flow, as an optimal experience, can elicit positive emotions and enhance subjective well-being ([8]; [60]). These studies collectively suggest that flow fosters psychological growth by generating pleasurable emotions. In line with this, the present study shows that when EFL learners experience flow in a language task, they can experience intrinsic satisfaction and academic achievement in the learning process. These positive emotional experiences help enhance their self-efficacy and sense of hope, which, in turn, increases the overall level of PsyCap. Meanwhile, neuroscientific evidence also suggests that the flow can activate the dopamine reward system and other parts of the brain, which in turn enhances the individual’s sense of pleasure and motivation ([70], [69]) and facilitates the generation of positive resources such as PsyCap. In addition, it has been shown that humans naturally strive for psychological growth, internalization, and well-being, driven by the satisfaction of three basic needs, i.e., autonomy, competence, and relatedness ([18]). This view explains the relationship between EFL learners’ flow and PsyCap, as flow stimulates interest and intrinsic motivation, satisfying autonomy and competence needs, which in turn fosters PsyCap and obtains a sense of intrinsic satisfaction.

### 5.2. The Mediating Role of Academic Efficacy

Academic efficacy—learners’ task-specific competence beliefs ([63]; [64])—is a key “built resource” in BBT ([24]). Flow broadens learners’ engagement (e.g., focused task participation; [26]), providing competence evidence that strengthens academic efficacy ([33]). As academic efficacy aligns with PsyCap’s self-efficacy component ([47]), this process embodies BBT: flow builds academic efficacy, which consolidates into enduring PsyCap.

In this study, the statistical results revealed that flow significantly and positively predicted the academic efficacy of EFL learners, which, in turn, impacted their PsyCap. Specifically, flow enhances academic efficacy by providing learners with positive learning experiences and reducing stress and anxiety in the learning process, thereby promoting the development of PsyCap. This finding underscores the significance of flow in EFL learning. Optimizing teaching design and learning tasks to help students achieve a state of flow can directly boost their academic efficacy and provide robust support for the accumulation of PsyCap. This result is consistent with the findings of multiple studies, including those by [2] ([2]), who demonstrated that academic efficacy is a critical factor in enhancing students’ intrinsic motivation and overall academic performance, and [10] ([10]), who highlighted the role of flow in enhancing academic self-efficacy.

When EFL learners experience flow, they are more likely to feel confident in their academic abilities. This increased confidence enhances their academic efficacy, which in turn reduces their academic burnout. As demonstrated by [2] ([2]), and [64] ([64]), higher levels of academic efficacy not only directly contribute to increased PsyCap but also foster a positive attitude towards learning, enhance intrinsic motivation, and promote resilience in the face of setbacks. Additionally, this finding aligns with the BBT ([24]), which suggests that positive emotional states, such as flow, can expand cognitive resources and build PsyCap.

In summary, academic efficacy is a key mediator between flow and PsyCap. This is consistent with the study’s conceptual framework, which draws on the BBT and emphasizes the role of flow in enhancing PsyCap through positive emotional experiences and intrinsic motivation ([13]; [24]).

### 5.3. The Chain Mediation Effect of FLCA and Academic Efficacy

FLCA—situation-specific tension in EFL contexts ([35]; [38])—reflects BBT’s ([24]) view of negative emotions as “narrowing” forces: it diverts learners’ focus to threats (e.g., fear of mistakes) and limits engagement ([68]), blocking PsyCap development. However, flow ([13])—a positive emotional state—undoes this narrowing via BBT’s “undoing effect”: it reduces FLCA by shifting attention to task immersion ([50]), removing barriers for subsequent resource building.

This study also explored the interplay between flow, FLCA, academic efficacy, and PsyCap. The results revealed that FLCA and academic efficacy play a significant chain-mediating role in the relationship between flow and PsyCap. Specifically, flow indirectly enhances learners’ PsyCap by first reducing FLCA, which in turn boosts academic efficacy.

FLCA is a common emotional challenge faced by many learners, often leading to negative outcomes such as reduced participation, lower motivation, and impaired performance ([31]). However, the findings of this study indicate that flow can mitigate these adverse effects by creating a more engaging and less stressful learning environment. When learners are in a state of flow, they are more likely to feel immersed and focused on the learning task, thereby reducing anxiety ([10]). This anxiety reduction then allows learners to develop a stronger sense of academic efficacy, as they feel more capable of overcoming academic challenges.

EFL learners with higher academic efficacy are more likely to maintain a positive attitude towards learning, persist in the face of difficulties, and engage more deeply with learning materials. This positive cycle of reduced anxiety and increased efficacy ultimately contributes to the accumulation of PsyCap, a set of qualities including self-efficacy, hope, resilience, and optimism.

The chain-mediation effect of FLCA and academic efficacy underscores the importance of addressing anxiety and fostering academic efficacy. Educators can create a more supportive learning environment that promotes the development of PsyCap. This, in turn, can lead to improved educational outcomes and enhanced overall well-being for EFL learners.

In summary, the study demonstrates that the relationship between flow and PsyCap is mediated by a chain of effects involving FLCA and academic efficacy. By understanding and leveraging these mechanisms, educators can design more effective teaching strategies to optimize the learning experience and support the holistic development of EFL learners.

### 5.4. Limitations and Future Research

Although the present study provides empirical evidence for the relationships among flow, FLCA, academic efficacy, and PsyCap in EFL learning, several limitations should be acknowledged.

Firstly, although all participants were university EFL learners taking compulsory English courses, individual learner characteristics (e.g., differences in learning engagement or academic self-beliefs) may still influence the observed relationships, which limits the generalizability of the findings. Future studies could consider more diverse learner profiles to capture these individual differences better.

Secondly, the sample selected in this study focuses on college learners, which does not adequately represent diverse cultural backgrounds or age stages, and subsequent studies can focus on other groups (e.g., secondary school students and adult education groups). Future studies may consider secondary school students, adult learners, or cross-cultural samples.

Thirdly, this study adopted a large sample cross-sectional approach. Although the results could test the association between variables, they could not show the causal direction of flow and PsyCap, which could be further verified by longitudinal tracking or experimental intervention. Future studies might use longitudinal or intervention designs to confirm the causal relationships posited in this model.

Finally, although a robustness test showed that differences in response formats (5-point vs. 7-point) did not substantially alter the direction or magnitude of the structural paths, minor scale-related inconsistencies may still exist in the current study. Future studies may benefit from using unified response formats or applying scale-harmonization techniques to minimize potential measurement bias further.

## 6. Conclusions

Focusing on the positive impact of EFL learners’ flow experience on their PsyCap, this study explores, theoretically, the mediating mechanisms of FLCA and academic efficacy, and examines their dual mediation in the EFL classroom. This study deepens the exploration of the complexity of the interaction between emotion and cognition in the process of EFL learning and helps understand how EFL learners can leverage the short-term pleasure of flow to promote the accumulation of PsyCap as a positive resource to sustain long-term persistence. At the same time, this study integrates interdisciplinary perspectives. It applies positive psychology to EFL learning scenarios, providing empirical evidence for expanding the BBT and for constructing a framework of EFL learning that emphasizes both “competence development” and “psychological health”.

In a practical sense, this study provides important insights for optimizing EFL teaching strategies. First, to foster students’ flow—as a critical pathway to developing PsyCap—EFL teachers can assess proficiency level by assigning tasks (e.g., sentence filling, vocabulary matching), and then tailor task difficulty to achieve a balance between challenge and skill. This balance not only sustains engagement but also strengthens learners’ sense of efficacy and optimism. Second, design flow-conducive environments—such as integrating gamified learning, cooperative goal-setting, and peer-supported activities—can mitigate FLCA. A balanced supportive–challenging environment encourages students to take academic risks, thereby facilitating the internalization of academic efficacy and the growth of resilience. These strategies collectively contribute to a virtuous cycle where positive emotions, reduced anxiety and burnout, and enhanced self-beliefs support sustainable language learning outcomes.

## Figures and Tables

**Figure 1 behavsci-15-01703-f001:**
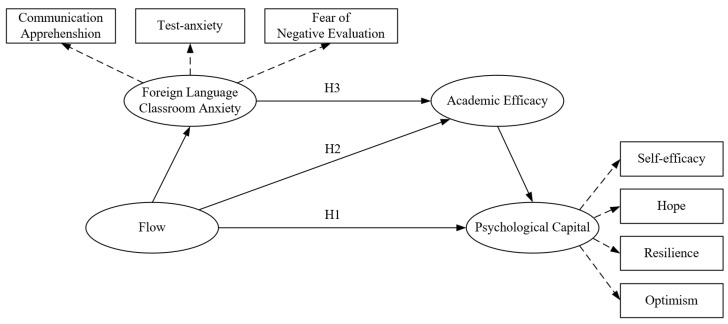
Hypothetical model.

**Figure 2 behavsci-15-01703-f002:**
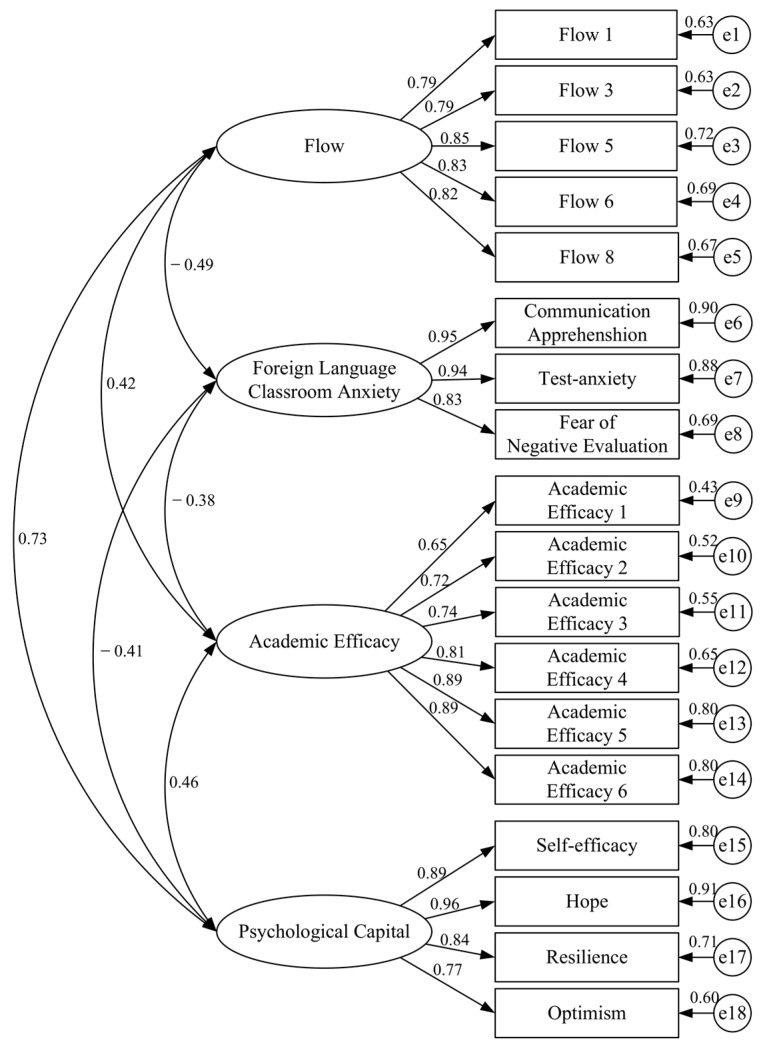
Measurement model.

**Figure 3 behavsci-15-01703-f003:**
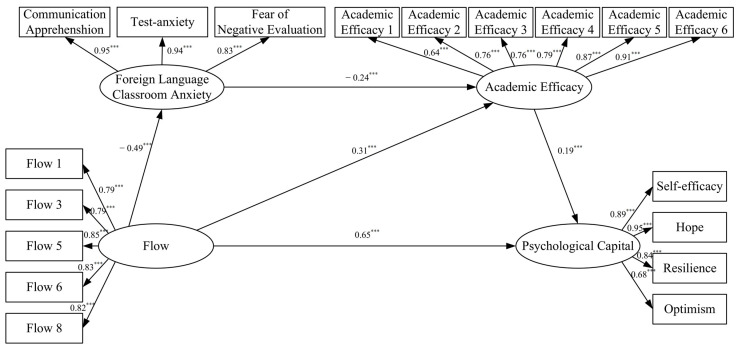
Chain mediation model path diagram. *Note*. *** *p* < 0.001.

**Table 1 behavsci-15-01703-t001:** Results of descriptive statistics and correlation analysis.

	*M*	*SD*	1	2	3	4
1. Flow	21.986	5.148	1			
2. FLCA	98.700	18.297	−0.456 ***	1		
3. Academic efficacy	27.350	8.279	0.405 ***	−0.370 ***	1	
4. PsyCap	82.270	12.987	0.672 ***	−0.396 ***	0.456 ***	1

*Note*. *** *p* < 0.001.

**Table 2 behavsci-15-01703-t002:** Mediation effect.

Paths	Effect Value	*SE*	95%CI		*p*
			Lower	Upper	
Flow → PsyCap (direct effect)	0.648	0.039	0.569	0.722	0.000
Flow→Academic Efficacy → PsyCap	0.059	0.012	0.036	0.084	0.000
Flow→FLCA→ Academic Efficacy→PsyCap	0.023	0.005	0.014	0.034	0.000
Total indirect effect	0.081	0.015	0.053	0.111	0.000
Total effect	0.729	0.029	0.668	0.784	0.000

## Data Availability

The datasets generated for this study are fully available upon reasonable request to the first and corresponding author.

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
