# Peer review of "The Impact of Flow on University EFL Learners’ Psychological Capital: Insights from Positive Psychology"

_behavsci, 2025, doi:10.3390/bs15121703_

Round 1
Reviewer 1 Report
Comments and Suggestions for Authors
Review result-BS
This study explored the impact of flow on Chinese university EFL learners’ psychological capital 10
(PsyCap), as well as the mediating roles of foreign language classroom anxiety (FLCA) and academic
Efficacy. It dealt with an underresearched topic—flow and had a large sample size, revealing very interesting findings. It has the potential to be published. Yet, it has many problems, which should be addressed before it is resubmitted for consideration.
Title: It’s better to specify what type of EFL learners, e.g., young learners/ middle school/ …
Abstract: Abstract normally reports what was done, how it was done and what was found, etc., not what aims to do.
Introduction
The writers introduce and describe flow, PsyCap, anxiety and efficacy. Since much research has been done on them and they have proved to be beneficial to L2 learning. Then, why should the writers still research them? Please do clearly state the rationales for the research.
Conceptual and theoretical framework, literature review, and hypotheses
This part is rather messy and needs extensive revision.
1) Does the broaden-and-build theory only account for flow and PsyCap? How about other emotions like anxiety and hope? How does the theory explain the relationships between the studied variables? Does it account for efficacy as well? In BBT, hope is a positive emotion, yet it is a component of PsyCAp in the study. How to explain this and its relation with flow?
2) For each variable (i.e., flow, anxiety, etc.), the review is simple and superficial, failing to illustrate what has been researched and why it is still worth research.
3) Some variables are defined, some are not. E.g., there are no definitions for flow and academic, but definitions are indeed necessary.
4) Since PsyCap covers self-efficacy, how is this efficacy different from academic efficacy as an independent variable?
5) Overlapping of 2.2.2 and 2.2.3 on academic efficacy.
6) Why is academic efficacy treated as a mediator, while the role of anxiety is unclear?
7) There is no ‘significantly’ in all the hypotheses, why? It’s often the case that many independent variables affect the same dependent variable, but just not significantly. So, what is the significance of specifically examining whether A affects/mediates B?
8) Generally speaking, hypotheses are predictions/possible answers to research questions. To me, I had never read a research article starting with hypotheses and then formulating RQs. It’s really strange.
9) Prior to each hypothesis, it’s better to state why it is necessary to do research to confirm it.
10) It’s better to show the hypotheses, e.g., H1/H2/H4 of the relationships in the hypothesized model.
Methods
- Participants: More info. is needed to better understand and interpret the findings, e.g., what university, places of universities, what major, year of study, English courses, English level, etc. how were they sampled?
Discussion
- How can the findings be explained by BBT?
- Do participants characteristics affect the findings? For example, some participants might not take any English courses and have lower self-efficacy thank those who were still learning English. This is why it’s important to provide more participant info. earlier in Methods.
Author Response
- This study explored the impact of flow on Chinese university EFL learners’ psychological capital (PsyCap), as well as the mediating roles of foreign language classroom anxiety (FLCA) and academic Efficacy. It dealt with an underresearched topic—flow and had a large sample size, revealing very interesting findings. It has the potential to be published. Yet, it has many problems, which should be addressed before it is resubmitted for consideration.
Response to Comments: We are grateful to the reviewer for recognizing the novelty of our work and the potential contribution of examining flow in relation to psychological capital (PsyCap) among EFL learners, and for noting the strength of our large sample. We welcome the reviewer’s overall assessment and take seriously the comment that the manuscript currently has “many problems” that need addressing. We appreciate the invaluable time and insights contributed by the two expert Reviewers. We have carefully considered all the feedback provided and have thoroughly revised and resubmitted our manuscript accordingly.
In the following sections, we will address each point raised, providing detailed explanations and justifications for the changes we have made. We understand the importance of a detailed and high-quality response letter that parallels the points raised by both the Reviewers and you, and we have formatted our responses to facilitate the review process.
We hope that our revisions meet your expectations and look forward to your feedback.
- Title: It’s better to specify what type of EFL learners, e.g., young learners/ middle school/ …
Response to Comments: We sincerely thank the reviewer for this helpful suggestion. We have clarified the characteristics of the participants in the revised manuscript. Specifically, the participants were Chinese university-level EFL learners, recruited from six comprehensive universities in mainland China. They were undergraduate students enrolled in various majors, all of whom were taking compulsory English courses as part of their degree programs.
The revised title is “The Impact of Flow on University EFL Learners’ Psychological Capital: Insights from Positive Psychology”. (p1, LN 2-3)
- Abstract: Abstract normally reports what was done, how it was done and what was found, etc., not what aims to do.
Response to Comments: We appreciate your insightful comment. In light of your observation and a parallel suggestion from Reviewer 2 to clarify the definition of “flow”, we have revised the abstract.
The relevant section now reads:
’ (p1, LN 12-14)
Introduction
- The writers introduce and describe flow, PsyCap, anxiety and efficacy. Since much research has been done on them and they have proved to be beneficial to L2 learning. Then, why should the writers still research them? Please do clearly state the rationales for the research.
Response to Comments: Thank you for your valuable comment. Although these variables have been separately investigated in the L2 field, empirical studies exploring their integrated mechanism remain limited. In the revised manuscript, we have expanded the Introduction to provide a more integrated rationale for the study. Specifically, we clarify that although flow, PsyCap, anxiety, and academic efficacy have been widely examined, empirical research rarely investigates how these positive and negative states jointly contribute to PsyCap development, nor how these mechanisms operate through chain-mediated emotional and motivational pathways within EFL contexts.
“To summarize, many studies have demonstrated that flow, PsyCap, anxiety, and academic efficacy are beneficial to L2 learning. However, little attention has been paid to how these positive and negative states jointly shape learners’ PsyCap development. In other words, empirical studies exploring their integrated psychological mechanism remain limited. Prior studies have mainly examined dyadic relationships (e.g., flow–motivation, anxiety–performance), but few have tested the chain mediation among these positive and negative psychological factors in EFL contexts. Examining the chain mediation allows us to uncover how flow exerts its influence through emotional and motivational processes, thereby clarifying the underlying mechanism through which short-term experiences translate into long-term psychological resources such as PsyCap. Based on this, this study integrates academic efficacy and FLCA into a single model to clarify how flow can foster enduring PsyCap. This framework addresses a gap in understanding how momentary experiences shape long-term psychological growth in EFL learning within Chinese cultural contexts. This research perspective not only helps to enrich the theoretical foundation in the field of flow and PsyCap, but also provides new insights into educational practices that can effectively enhance EFL learners’ PsyCap and academic achievement by facilitating flow and reducing anxiety.” (p 3, LN 99-115)
Conceptual and theoretical framework, literature review, and hypotheses
- This part is rather messy and needs extensive revision.
1) Does the broaden-and-build theory only account for flow and PsyCap? How about other emotions like anxiety and hope? How does the theory explain the relationships between the studied variables? Does it account for efficacy as well? In BBT, hope is a positive emotion, yet it is a component of PsyCap in the study. How to explain this and its relation with flow?
Response to Comments: Thank you for this insightful comment. We agree that a clearer theoretical explanation is needed regarding how the BBT (Fredrickson, 2001) accounts for all the studied variables. According to BBT, positive emotions such as hope, and efficacy-related confidence broaden individuals’ thought and action, whereas negative emotions like anxiety narrow attention and restrict exploration.
In our model, flow represents a transient positive state that broadens learners’ attention and engagement in language tasks. This broadened attention facilitates academic efficacy, since learners experience mastery and competence during the positive state (Hayat et al., 2020). At the same time, these positive experiences help attenuate foreign language classroom anxiety, a debilitating negative emotion that restricts learners' cognitive resources (Teimouri et al., 2019). Hence, the reduction of anxiety and the increase of efficacy both exemplify the “broaden-and-build” cycle in action.
Regarding hope, we recognize that hope plays dual theoretical roles. Within BBT, hope is a positive emotion characterized by goal-directed energy and perceived pathways (Fredrickson, 2001). Within psychological capital (PsyCap), it functions as a trait-like resource reflecting one’s enduring capacity to set and pursue goals (Luthans et al., 2007). Thus, in our framework, flow acts as an antecedent positive experience that triggers momentary hope, which—when repeatedly experienced—accumulates into the trait-like hope component of PsyCap. This interpretation aligns with longitudinal findings that repeated flow experiences foster enduring psychological resources such as resilience and optimism. (Mao et al., 2024; Wu et al., 2021).
Therefore, the BBT framework not only accounts for flow and PsyCap but also explains how positive emotions (flow, academic efficacy) dynamically counterbalance negative emotions (anxiety) and jointly contribute to the accumulation of PsyCap. We have revised the original paragraph in the theoretical foundation section to clarify this conceptual alignment.
“This study is based on the broaden-and-build theory (BBT), which argues that positive emotional experiences expand an individual’s thinking and actions. Positive emotions can initiate cycles of beneficial emotions and behaviors through dynamic psychological processes, significantly influencing long-term emotional health and well-being (Fredrickson & Joiner, 2018). This contributes to the development of enduring personal resources, including physical, intellectual, social, and psychological assets (Fredrickson, 2001).
Within this framework, flow functions as a transient positive state that broadens learners’ attention and engagement, enhancing their sense of mastery and academic efficacy (Hayat et al., 2020). At the same time, these positive experiences help attenuate foreign language classroom anxiety, a debilitating negative emotion that restricts learners’ cognitive resources (Teimouri et al., 2019). Furthermore, hope—as defined in the BBT—is a positive emotion that arises in challenging situations and motivates individuals to pursue more favorable outcomes (Fredrickson, 2013). When repeatedly experienced through flow, this momentary hope can gradually evolve into the trait-like hope component of psychological capital (Luthans et al., 2007).
Therefore, the BBT offers an integrative explanation for how positive emotions (flow, academic efficacy) and negative emotions (anxiety) interact in EFL learning to influence learners’ PsyCap development. This theoretical framework not only reveals these dynamic relationships but also guides the optimization of EFL instruction by creating contexts that foster flow and support the development of psychological resources”(p 3 , LN 118-139)
2) For each variable (i.e., flow, anxiety, etc.), the review is simple and superficial, failing to illustrate what has been researched and why it is still worth research.
Response to Comments:
Thank you for pointing out this important issue. We appreciate the reviewer’s suggestion to strengthen the literature review. In the revised manuscript, we have expanded the literature review to fill the existing research gaps and enhance the value of this study.
“Flow is a mental state in which a person is fully immersed in an activity, experiencing energized focus, full involvement, and enjoyment (Csikszentmihalyi & Nakamura, 1990). In EFL research, flow has been viewed as a relevant variable for predicting students’ interest, teachers’ teaching effectiveness, and the creation of an optimal learning state (Jia et al., 2025).
Empirical studies have shown that flow can improve learning efficiency and engagement by driving intrinsic motivation and alleviating external pressure (Gao et al., 2025). Furthermore, creating activities that promote flow can help improve EFL learners’ input skills (i.e., reading), enhance their sense of efficacy (Shahian et al., 2017), help them clarify their learning goals for output skills (i.e., writing), and improve their resistance to interruptions (Liu et al., 2022). In addition, flow promotes intrinsic motivation and sustained cognitive involvement, creating learning conditions that support deeper processing and greater academic persistence. Evidence from weekly diary studies and scale-validation work further indicates that flow strengthens learners’ self-beliefs and supports optimal academic functioning (Jia et al., 2024; Jia et al., 2025). However, although studies have linked flow to improved motivation and emotional well-being, much less is known about the emotional mechanisms through which flow contributes to long-term psychological development. This gap highlights the need to examine flow within a more integrative emotional and motivational framework.” (p 3, LN 142-148; p 4, 149-160)
2.1.2. Psychological capital (PsyCap)
“As one of the key psychological resources supporting optimal learning experiences, positive psychological resources have received growing attention in EFL research. Among these resources, psychological capital (PsyCap) is regarded as a core component. PsyCap, proposed by Luthans (2007), refers to a set of positive psychological states—self-efficacy, hope, optimism, and resilience—that individuals exhibit during their growth and development, and it includes a range of psychological resources that can be developed and nurtured. In educational settings, abundant PsyCap has been associated with greater resilience, enhanced motivation, and improved emotional well-being, making it a meaningful construct for understanding learners’ long-term psychological functioning.
A growing body of empirical work has shown that PsyCap benefits EFL learners in multiple ways. Higher levels of PsyCap predict greater behavioral, emotional, and cognitive engagement (Derakhshan & Noughabi, 2024; King & Caleon, 2021) and are linked to higher academic enjoyment and lower academic boredom (Wu & Kang, 2023). PsyCap also enhances learners’ adaptability and coping strategies when dealing with stressors in language learning (Lin, 2020). Moreover, PsyCap contributes to learners’ persistence, motivation, and willingness to engage with challenging tasks (Khajavy et al., 2019). However, little is known about how PsyCap develops through emotional processes, particularly when positive states (e.g., flow) and negative emotions (e.g., anxiety) are considered together. This limitation underscores the need for a model that integrates emotional and cognitive variables to clarify the developmental mechanisms of PsyCap in EFL contexts.” (p 4, LN 162-183)
“Anxiety is a situation-specific emotional reaction characterized by apprehension, tension, and worry over perceived threats to one’s self-worth or performance (Horwitz et al., 1986). Within formal educational settings, anxiety typically surfaces when learners anticipate negative evaluation, fear failure, or doubt their ability to meet academic demands (Zeidner, 1998). In the foreign-language classroom, these feelings are amplified by the need to communicate in an unfamiliar linguistic code, giving rise to what is known as foreign language classroom anxiety (FLCA). FLCA refers to the tension and worry learners experience in foreign language learning situations, particularly during speaking and listening activities (Jin et al., 2024).” (p 5, LN 224-233).
3) Some variables are defined, some are not. E.g., there are no definitions for flow and academic, but definitions are indeed necessary.
Response to Comments:
Thank you for this helpful comment. We agree that clear conceptual definitions are essential for theoretical consistency, and we have revised them in the revised draft.
“Flow is a mental state in which a person is fully immersed in an activity, experiencing energized focus, full involvement, and enjoyment (Csikszentmihalyi & Nakamura, 1990).” (p 3, LN 142-144)
“In the context of English as a Foreign Language (EFL) learning, "academic" is often used to describe language learning that occurs within educational institutions, as opposed to naturalistic or informal language acquisition. It includes classroom-based instruction, teacher-assigned tasks, assessments, and students’ perceptions of their learning experiences in these settings (Artino, 2012; Schaufeli et al., 2002). For example, academic efficacy refers to students' beliefs in their ability to succeed in academic-specific tasks, such as completing assignments, passing exams, or mastering course content. These beliefs are shaped by academic feedback, prior performance, and instructional support, and they directly influence students’ motivation, persistence, and emotional responses in educational environments (Pajares & Schunk, 2001; Zimmerman, 2000).” (p 4, LN 185-195)
4) Since PsyCap covers self-efficacy, how is this efficacy different from academic efficacy as an independent variable?
Response to Comments:
We appreciate this conceptual clarification question. While PsyCap does include self-efficacy as one of its four core components (Luthans et al., 2007), the Academic Efficacy Scale (Schaufeli et al., 2002) we employ in the current study measures a distinct, context-specific construct rooted in Bandura's self-efficacy theory (Schunk & DiBenedetto, 2022). PsyCap’s self-efficacy subscale assesses general confidence in handling diverse challenges, whereas academic efficacy specifically captures learners' beliefs about their capacity to manage academic tasks and challenges.
5) Overlapping of 2.2.2 and 2.2.3 on academic efficacy.
Response to Comments:
We appreciate your comment that sections 2.2.2 (simple mediation) and 2.2.3 (chain mediation) redundantly discuss academic efficacy, creating conceptual overlap. Initially, we separated them to emphasize distinct theoretical mechanisms, but this led to repetitive exposition of the same construct. Following Hair et al.'s (2010) guidance on model clarity and Sathyanarayana & Mohanasundaram's (2024) parsimony standards, we will consolidate all conceptual development of academic efficacy—its theoretical roots (Schunk & DiBenedetto, 2022), measurement (Schaufeli et al., 2002), and relevance to PsyCap (Luthans et al., 2007)—into a unified preliminary subsection. Sections 2.2.2 and 2.2.3 will then focus exclusively on articulating the hypothesized pathways without redefining the construct, ensuring each section serves a unique analytical purpose.
6) Why is academic efficacy treated as a mediator, while the role of anxiety is unclear?
Response to Comments:
We appreciate your insightful critique. The distinction stems from their fundamentally different functions within the broaden-and-build framework (Fredrickson, 2001) in which academic efficacy is a positive resource-building variable that directly constitutes PsyCap (Luthans et al., 2007; Schunk & DiBenedetto, 2022), making it a clear simple mediator (Flow→Efficacy→PsyCap). Conversely, FLCA is a negative emotional barrier that flow must reduce before efficacy can be enhanced (Horwitz et al., 1986; Jin et al., 2024), positioning it as a sequential component in a chain (Flow→FLCA→Efficacy→PsyCap) rather than a direct PsyCap builder. We recognize our H3 wording (“chain mediating role of FLCA and academic efficacy”) obscured this nuance by implying FLCA directly mediates, when its function is conditional—neutralizing anxiety merely enables the efficacy pathway. We will revise by: (1) rephrasing H3 to specify the sequential path, (2) labeling paths in Figure 1, and (3) clarifying in the Discussion that FLCA operates as a disabler neutralized by flow, not an enabler like efficacy, thereby accurately representing the differential mechanisms of resource accumulation versus barrier removal.
“Hypothesis 3: FLCA and academic efficacy sequentially and significantly mediate the relationship between flow and PsyCap. Specifically, flow first significantly alleviates FLCA; this anxiety reduction subsequently significantly strengthens academic efficacy, which in turn significantly elevates PsyCap.”(p 7, LN 322-325)
7) There is no ‘significantly’ in all the hypotheses, why? It’s often the case that many independent variables affect the same dependent variable, but just not significantly. So, what is the significance of specifically examining whether A affects/mediates B?
Response to Comments: Thank you very much for this insightful comment.
The revised versions read as follows:
“Hypothesis 1: Learners’ flow during EFL learning can significantly positively predict their PsyCap level.”(p 6, LN 270-271)
“Hypothesis 2: Academic efficacy significantly mediates the relationship between flow and PsyCap, that is, flow significantly improves academic efficacy, which in turn significantly enhances an individual’s PsyCap.”(p 6, LN 297-299)
“Hypothesis 3: FLCA and academic efficacy sequentially and significantly mediate the relationship between flow and PsyCap. Specifically, flow first significantly alleviates FLCA; this anxiety reduction subsequently significantly strengthens academic efficacy, which in turn significantly elevates PsyCap.”(p 7, LN 322-325)
8) Generally speaking, hypotheses are predictions/possible answers to research questions. To me, I had never read a research article starting with hypotheses and then formulating RQs. It’s really strange.
Response to Comments:
Thank you for your feedback. We referred to similar published papers and found that indeed RQs do not appear after hypotheses. Therefore, we have deleted the research questions based on your comments.
9) Prior to each hypothesis, it’s better to state why it is necessary to do research to confirm it.
Response to Comments:
We appreciate this constructive suggestion. We have made the following revisions:
When laying the groundwork for H1, we have revised: “It can be concluded that many studies have highlighted the significant positive correlations between flow and the individual components of PsyCap. However, few have examined the link between flow and PsyCap as a whole construct, which is essential for learners’ sustained engagement and success in EFL. Given the complexity of this relationship, it is also essential to consider the cross-cultural cognitive challenges specific to EFL contexts, which may influence the pattern of correlation between the two.” (p 6,LN 264-269)
For H2, we have revised: “Flow encourages students to adopt more effective learning strategies, such as active learning and metacognitive monitoring. Optimizing these strategies helps students complete learning tasks more efficiently, thereby improving academic efficacy (Bembenutty et al., 2024). As an essential component of flow, autonomy or the freedom of individuals to arrange activities has been repeatedly found to increase positive affect and motivation (Mao et al., 2020). Experiencing flow can temporarily enhance students’ confidence and reduce self-consciousness during academic tasks, thereby reinforcing their academic efficacy (Csikszentmihalyi et al., 1988).”(p 6,LN 289-296)
Regarding H3, we have revised: “FLCA may weaken students’ efficacy and motivation to learn (Shang & Ma, 2024), which in turn may further reduce students’ PsyCap, and learners with high PsyCap tend to have higher self-efficacy, stronger hope, greater resilience, and more optimistic attitudes (Newman et al., 2014), and these traits help reduce FLCA. Students who experience flow more frequently in daily life are more likely to have higher self-esteem and lower anxiety levels. Individuals who experience flow more often across domains also tend to have fewer depressive symptoms and less emotional exhaustion (Mosing et al., 2018).” (p 7,LN 315-322)
10) It’s better to show the hypotheses, e.g., H1/H2/H4 of the relationships in the hypothesized model.
Response to Comments:
Thank you for this valuable suggestion. In the revised version, we have labeled the corresponding paths in the hypothesized model (Figure 1).
Methods
- Participants: More info. is needed to better understand and interpret the findings, e.g., what university, places of universities, what major, year of study, English courses, English level, etc. how were they sampled?
Response to Comments: We sincerely thank the reviewer for pointing out the need for greater detail regarding the participants. We have substantially revised the Participants and Procedure section to provide a fuller description of the sample background and recruitment process. Specifically, we now clarify that participants were undergraduate students majoring in English or other non-English disciplines enrolled in required English courses at six public universities located in eastern, central, and southwestern China. Their English proficiency ranged from CEFR B1 to B2, as determined by their college placement tests.
We also added information about sampling procedures. Participants were recruited through convenience sampling with the assistance of course instructors who distributed the survey link in class. Participation was entirely voluntary.
“A total of 1,669 undergraduate EFL learners from six public universities across eastern, central, and southwestern China participated in this study. Both English and non-English majors were included, all of whom were enrolled in compulsory English courses. The participants were recruited through convenience sampling (Landers & Behrend, 2015), and instructors distributed the survey QR code during class time.”(p 7, LN 333-339)
Landers, R. N., & Behrend, T. S. (2015). An inconvenient truth: Arbitrary distinctions between organizational, Mechanical Turk, and other convenience samples. Industrial and Organizational Psychology, 8(2), 142–164. https://doi.org/10.1017/iop.2015.13
Discussion
- How can the findings be explained by BBT?
Response to Comments:
Thank you for this insightful question. In the revised Discussion, we have elaborated on how the present findings can be interpreted through the lens of BBT (Fredrickson, 2001).
“The results showed that EFL learners’ flow significantly and positively predicted their PsyCap, further supporting the BBT's view that positive emotions broaden individuals’ momentary thought and action repertoires and build enduring psychological resources (Fredrickson, 2001). Flow exemplifies a positive state in which learners’ flow helps regulate emotions and enhance engagement, and, over time, these positive emotional and cognitive expansions contribute to building more stable psychological resources, reflected in the higher levels of PsyCap observed in this study.” (p 11, LN 471-477)
“Academic efficacy—learners’ task-specific competence beliefs (Schaufeli et al., 2002; Schunk & DiBenedetto, 2022)—is a key "built resource" in BBT (Fredrickson, 2001). Flow broadens learners’ engagement (e.g., focused task participation; Fredrickson & Joiner, 2018), providing competence evidence that strengthens academic efficacy (Hayat et al., 2020). As academic efficacy aligns with PsyCap’s self-efficacy component (Luthans et al., 2007), this process embodies BBT: flow builds academic efficacy, which consolidates into enduring PsyCap.” (p 12, LN 497-503)
“FLCA—situation-specific tension in EFL contexts (Horwitz et al., 1986; Jin et al., 2024)—reflects BBT’s (Fredrickson, 2001) view of negative emotions as "narrowing" forces: it diverts learners’ focus to threats (e.g., fear of mistakes) and limits engagement (Teimouri et al., 1997), blocking PsyCap development. However, flow (Csikszentmihalyi, 1975)—a positive emotional state—undoes this narrowing via BBT’s "undoing effect": it reduces FLCA by shifting attention to task immersion (Mao et al., 2020), removing barriers for subsequent resource building.”(p 12, LN 530-532; p 13, LN 533-536)
- Do participants characteristics affect the findings? For example, some participants might not take any English courses and have lower self-efficacy thank those who were still learning English. This is why it’s important to provide more participant info. earlier in Methods.
Response to Comments:
Thank you for this helpful comment. In fact, all participants in this study were university EFL learners, as English courses are compulsory in Chinese higher education. We also acknowledge that individual learner characteristics may influence the findings, which represents a limitation of the present study. This point has now been incorporated into the Limitations and Future Research section.
“Firstly, although all participants were university EFL learners taking compulsory English courses, individual learner characteristics (e.g., differences in learning engagement or academic self-beliefs) may still influence the observed relationships, which limits the generalizability of the findings. Future studies could consider more diverse learner profiles to capture these individual differences better.” (p 13, LN 569-573)
Reviewer 2 Report
Comments and Suggestions for Authors
Abstract:
- No background information mentioned. The author will need to write 1-2 sentences to describe why it is important to study the flow impact of flow on EFL learners' psychological capital.
- What does "flow" mean in the context of EFL learning? A brief explanation of what "flow" refers to in the context of EFL learning would make the abstract clearer.
- Although the sample size (i.e., 1,611 EFL learners) from six universities is large, a concise description of the participants (e.g., proficiency levels, L1, and types of EFL courses) is needed.
- In the methods of the abstract, the type of questionnaire should be mentioned (e.g., is it a standard questionnaire or self-developed by the researchers?).
- Although the abstract states the main results as, “The results indicated that flow could significantly and positively predict the level of PsyCap. Moreover, academic efficacy played a mediating role between flow and PsyCap, while FLCA and academic efficacy formed a chain-mediated path. Specifically, flow indirectly enhanced learners’ PsyCap by reducing FLCA and promoting academic efficacy,” it doesn't include any key statistical values.
- The implications section of the abstract, such as "optimizing foreign language teaching strategies," is too broad.
Introduction:
- Although the introduction mentions that there are limited studies exploring language learners' formation of academic efficacy, the specific research gap that this study aims to fill needs to be clearly stated.
- There are some grammar issues like tense inconsistencies.
Literature Review:
- While the review covers a range of studies, it lacks a deep synthesis of those studies. It does not provide critical analysis of the methodologies, findings, and limitations of the reviewed empirical studies.
- The definitions of key concepts are sometimes repeated in multiple sections, making it difficult to read.
- There are issues with tense inconsistencies, too.
Method:
- There is no information on how the participants were recruited or selected. Were they randomly selected, or was a convenience sampling method used? What were the inclusion and exclusion criteria?
- Although the authors mentioned that ethical approval was obtained, the description of the ethical procedures is quite brief. What specific steps were taken to ensure informed consent forms were collected?
- The steps to adapt the Flow Scale is unclear. Although the authors mention that the scales have demonstrated strong validity and reliability in previous studies, this part needs more details.
- The authors should justify the use of different response scales, as the FKCAS is a 5-point Likert scale while the other scales in this study use a 7-point scale. How will the potential bias introduced by this discrepancy be addressed?
Results:
- A short justification for their choice of statistical analyses is needed. Why was CFA used? Why was SEM used for mediation analysis?
- The authors state that "All the standardized factor loadings for the indicators of constructs were above the cut-off value (0.40) and significant (p < .001)." This is unclear. Please report the specific factor loadings for each indicator.
- Although the fit index was reported, the authors need to indicate whether these values demonstrate an acceptable model fit based on established guidelines.
- The authors need to describe the path coefficients (e.g., whether the reported path coefficients are standardized or unstandardized).
- While specific indirect effects are given in Table 2, the total indirect effect is not explicitly stated.
Discussion:
- The authors restate the findings without providing a deep discussion. For example, they mentioned "flow can significantly and positively predict their PsyCap" but doesn't delve into the reasons why this might be the case, beyond a general reference to the broaden-and-build theory. While the broaden-and-build theory is mentioned, the first paragraph of the discussion does not explicitly explain how the study’s statistical findings confirm this theory.
- The discussion section cited a lot of studies without interpret the cited studies together with their main results. I feel like reading the literature review again.
- Although the author mentioned the limitation in their conclusion part. The discussion generally presents a positive view of the results without acknowledging any limitations or alternative explanations. A strong discussion section should critically evaluate the study's strengths and weaknesses and consider potential confounding factors. So, remove the limitation from the conclusion section and add a paragraph of limitations in the discussion.
Kindly have the paper polished by a native English speaker who is good at academic writing to improve the clarity and the tense inconsistencies.
Author Response
Abstract:
- No background information mentioned. The author will need to write 1-2 sentences to describe why it is important to study the flow impact of flow on EFL learners' psychological capital.
Response to Comments: Thank you for this valuable suggestion. We have added a concise background sentence emphasizing the theoretical and pedagogical significance of examining how flow contributes to EFL learners’ psychological capital from a positive psychology perspective.
Added text: “Many studies have shown that flow, psychological capital (PsyCap), anxiety, and academic efficacy play significant roles in EFL learning, yet little attention has been paid to how these positive and negative states jointly shape learners’ PsyCap. Grounded in the broaden-and-build theory, this study investigated how flow, a state of deep engagement and enjoyment in learning, affected EFL learners’ PsyCap” (p 1 , LN 10-14)
- What does “flow” mean in the context of EFL learning? A brief explanation of what “flow” refers to in the context of EFL learning would make the abstract clearer.
Response to Comments: We agree with the reviewer. A short definition of flow was incorporated in the revised abstract to clarify its meaning in the EFL context.
Added text: “’ (p1, LN 12-14)
- Although the sample size (i.e., 1,611 EFL learners) from six universities is large, a concise description of the participants (e.g., proficiency levels, L1, and types of EFL courses) is needed.
Response to Comments: We have revised the abstract to include the essential participant information: the number of learners, the number of universities, and the country of data collection. More detailed demographic and academic characteristics are presented in the “Participants and Procedure” section of the main text.
Added text: “ total of 1,611 EFL learners at the CEFR B1–B2 levels from six universities in China participated in the study.” (p 1, LN 14-15)
- In the methods of the abstract, the type of questionnaire should be mentioned (e.g., is it a standard questionnaire or self-developed by the researchers?).
Response to Comments: We appreciate this suggestion. The revised abstract now explicitly states that data were collected through validated self-developed questionnaires measuring flow, FLCA, academic efficacy, and PsyCap.
Added text: “Data were collected using validated questionnaires developed for this study that measured flow, foreign language classroom anxiety (FLCA), academic efficacy, and PsyCap, and analyzed using structural equation modeling (SEM) in AMOS.” (p1 , LN 15-18)
- Although the abstract states the main results as, “The results indicated that flow could significantly and positively predict the level of PsyCap. Moreover, academic efficacy played a mediating role between flow and PsyCap, while FLCA and academic efficacy formed a chain-mediated path. Specifically, flow indirectly enhanced learners’ PsyCap by reducing FLCA and promoting academic efficacy,” it doesn't include any key statistical values.
Response to Comments: Thank you for this helpful comment. We have revised the abstract to include key path coefficients and significance levels from the SEM analysis, ensuring that the reported findings are supported by quantitative evidence.
Added text:
“The results revealed that flow had a significant direct positive effect on PsyCap (β = 0.648, p < .001). Academic efficacy significantly mediated this relationship (β = 0.059, p < .001), and a significant chain-mediated path was observed through FLCA and academic efficacy (β = 0.023, p < .001). The total effect of flow on PsyCap was 0.729 (p < .001).” (p 1 , LN 18-21)
- The implications section of the abstract, such as “optimizing foreign language teaching strategies,” is too broad.
Response to Comments: We appreciate this comment. The implications have been refined to emphasize the theoretical and pedagogical relevance of fostering flow to enhance learners’ psychological resources, avoiding overly general statements.
Revised sentence:
“These findings provide new insights into educational practices that can effectively enhance EFL learners’ PsyCap and academic achievement by facilitating flow and reducing anxiety.” (p 1 , LN 21-23)
Introduction:
- Although the introduction mentions that there are limited studies exploring language learners' formation of academic efficacy, the specific research gap that this study aims to fill needs to be clearly stated.
Response to Comments:
We appreciate this comment. While we cited Artino (2012) to note limited research on academic efficacy formation, we failed to clearly articulate the specific gap our study addresses: no prior research has tested academic efficacy as a mediator in the flow-PsyCap relationship within EFL contexts. Although Piniel and Albert (2017) demonstrated flow enhances self-efficacy, and Luthans et al. (2007) identify self-efficacy as a core PsyCap component, their integrated mechanistic link remains unexamined. Similarly, while FLCA impedes language learning (Horwitz et al., 1986; Jin et al., 2024) and relates to flow (Mao et al., 2020), its position as a barrier that flow must reduce to unlock efficacy gains has not been empirically modeled. We will revise the Introduction to explicitly state that our study uniquely fills this gap by testing how flow builds PsyCap through both direct resource-building (efficacy) and barrier-removal (anxiety) pathways, thereby advancing understanding of emotional-cognitive interactions in SLA.
“Despite growing interest in positive-psychology constructs, two critical gaps remain. First, no study has tested whether domain-specific academic efficacy functions as a mechanism through which flow experiences translate into higher-order psychological capital in EFL settings: a. Although Piniel & Albert (2017) linked flow to self-efficacy and Luthans et al. (2007) located efficacy at the core of PsyCap, their integrated causal chain has yet to be examined. Second, while FLCA is known to impede language learning (Horwitz et al., 1986; Jin et al., 2024) and covary with flow (Mao et al., 2020), its sequential role—as a barrier that must be reduced by flow before academic efficacy and PsyCap can accrue—has never been modelled. The present study addresses both lacunae by simultaneously evaluating (1) a direct resource-building path (flow → academic efficacy → PsyCap) and (2) a barrier-removal path (flow → FLCA → academic efficacy → PsyCap), thereby offering the first comprehensive test of how optimal experience fosters sustainable psychological resources in SLA.” (p 5, LN 209-221)
“Despite extensive research identifying Foreign Language Classroom Anxiety (FLCA) as a complex, context-specific construct (Gu et al., 2024; Horwitz et al., 1986; Jin et al., 2024) with detrimental effects on language learning engagement and achievement (Qiangfu, 2024; Xu & Xie, 2024), critical gaps remain: (a) unclear causal direction and interactive mechanisms between FLCA and flow in EFL contexts, despite flow’s potential to mitigate anxiety (Dewaele & Li, 2021); (b) no empirical testing of FLCA’s sequential mediating role alongside academic efficacy in the "flow→PsyCap" pathway; and (c) insufficient exploration of contextualized variations in FLCA’s impact (e.g., digital vs. in-person settings; Zeidner, 1998) and the mechanisms through which it constrains key psychological resources like academic efficacy and PsyCap.”(p 5, LN 244-251; P 6, LN 252-253)
- There are some grammar issues like tense inconsistencies.
Response to Comments: Thank you for pointing out the grammar and tense inconsistencies. We have carefully reviewed the entire manuscript and made comprehensive revisions to improve clarity, consistency, and academic writing style. Specifically, we addressed the following types of issues:
Tense inconsistencies: Ensured consistent use of present tense when describing general findings or established literature (e.g., “increase substantially” instead of “show a significant upward trend”) and past tense when describing our study procedures or results.
Sentence structure and parallelism: Revised complex or repetitive sentences for smoother readability (e.g., “focus on imparting knowledge and skills while paying little attention…”).
Word choice and clarity: Improved wording for conciseness and precision (e.g., “PsyCap is a core positive psychological resource that enhances…”).
These corrections have been applied throughout the manuscript to ensure grammatical accuracy and readability. For specific examples, please refer to the revised manuscript.
Literature Review:
- While the review covers a range of studies, it lacks a deep synthesis of those studies. It does not provide critical analysis of the methodologies, findings, and limitations of the reviewed empirical studies.
Response to Comments:
We appreciate your critique that our review lacked critical synthesis. In prioritizing comprehensiveness, we failed to integrate studies methodologically. We have revised by adding this text:
“To summarize, many studies have demonstrated that flow, PsyCap, anxiety, and academic efficacy are beneficial to L2 learning. However, little attention has been paid to how these positive and negative states jointly shape learners’ PsyCap development. In other words, empirical studies exploring their integrated psychological mechanism remain limited. Prior studies have mainly examined dyadic relationships (e.g., flow–motivation, anxiety–performance), but few have tested the chain mediation among these positive and negative psychological factors in EFL contexts. Examining the chain mediation allows us to uncover how flow exerts its influence through emotional and motivational processes, thereby clarifying the underlying mechanism through which short-term experiences translate into long-term psychological resources such as PsyCap. Based on this, this study integrates academic efficacy and FLCA into a single model to clarify how flow can foster enduring PsyCap. This framework addresses a gap in understanding how momentary experiences shape long-term psychological growth in EFL learning within Chinese cultural contexts. This research perspective not only helps to enrich the theoretical foundation in the field of flow and PsyCap, but also provides new insights into educational practices that can effectively enhance EFL learners’ PsyCap and academic achievement by facilitating flow and reducing anxiety.” (p 3, LN 99-115)
- The definitions of key concepts are sometimes repeated in multiple sections, making it difficult to read.
Response to Comments:
We appreciate this comment about definitional redundancy. In striving to ensure each section stood independently, we inadvertently repeated definitions of flow (Csikszentmihalyi, 1975), PsyCap (Luthans et al., 2007), academic efficacy (Schunk & DiBenedetto, 2022), and FLCA (Horwitz et al., 1986) across the Introduction and Literature Review, disrupting reading flow. To enhance clarity and streamline the narrative, we will consolidate all concept definitions into Section 2.2 (Constructs and related studies) as a single authoritative reference, with subsequent sections using consistent terminology without redefinition, thereby improving manuscript coherence and reader experience.
- There are issues with tense inconsistencies, too.
Response to Comments:
Thank you for your comment. We have systematically reviewed present tense for established constructs and prior findings (e.g., “FLCA impedes learning,” Horwitz et al., 1986), and past tense exclusively for our methodology (“We measured”) and results (“The results indicated”), eliminating all inconsistencies to meet scholarly writing conventions.
Method:
- There is no information on how the participants were recruited or selected. Were they randomly selected, or was a convenience sampling method used? What were the inclusion and exclusion criteria?
Response to Comments: We sincerely thank the reviewer for raising this important point. We have now revised the manuscript to provide a detailed description of the participant recruitment and selection process, including the sampling method, inclusion/exclusion criteria, and data cleaning procedures. The added text is as follows:
Revised Manuscript Text:
“A total of 1,669 undergraduate EFL learners from six public universities across eastern, central, and southwestern China participated in this study. Both English and non-English majors were included, all of whom were enrolled in compulsory English courses. The participants were recruited through convenience sampling (Landers & Behrend, 2015), and instructors distributed the survey QR code during class time. To ensure data quality, responses that failed to answer trap questions or showed identical answers were excluded from the analysis.”(p 7, LN 333-339)
- Although the authors mentioned that ethical approval was obtained, the description of the ethical procedures is quite brief. What specific steps were taken to ensure informed consent forms were collected?
Response to Comments: We thank the reviewer for highlighting the need for more detail regarding the ethical process. In the revised manuscript, we have expanded this section to clarify the ethical approval process and informed consent procedures. Before data collection, approval was obtained from the Research Ethics Committees of the participating universities. An informed consent statement was displayed on the first page of the online survey. Participants could only proceed to the questionnaire after confirming their voluntary agreement. No personally identifiable information was collected.
“Before data collection, ethical approval was obtained from the ethical review boards of all participating universities. Participants were provided with detailed information about the study’s purpose, procedures, data confidentiality, and their rights. It was explicitly stated that participation was entirely voluntary, that they could withdraw at any time without penalty, and that their decision would not affect their course grades or relationship with the instructors. Furthermore, we assured participants that there were no right or wrong answers to the questionnaire items. A QR code linked to the questionnaire was shared with the participants, who were invited to scan the code and complete the survey voluntarily.” (p 8, LN 343-351)
- The steps to adapt the Flow Scale is unclear. Although the authors mention that the scales have demonstrated strong validity and reliability in previous studies, this part needs more details.
Response to Comments: Thank you for pointing out that the adaptation procedure of the Flow Scale required further clarification. We have now revised this section to provide a more detailed and transparent description of the steps taken.
Specifically, we now clarify that we adopted the Chinese translation used in Mao et al. (2016), which has demonstrated satisfactory psychometric properties in multinational samples.
The revised manuscript now states:
“To ensure linguistic and conceptual equivalence, we adopted the Chinese translation of the scale used in Mao et al. (2016), which has been validated across American, Chinese, and Spanish samples (α = 0.77, 0.88, and 0.82, respectively; Mao et al., 2016; Bonaiuto et al., 2016; Jia et al., 2024). Based on the results of the factor analysis, item 2 (loading = 0.25), item 4 (loading = 0.24), and item 7 (loading = 0.21) were excluded from the analysis, as their factor loadings were below the recommended threshold of 0.40 (Hair et al., 2010). According to George and Mallery’s criteria (2003), the scale demonstrated excellent reliability, with a Cronbach’s alpha of 0.908 in the current sample.” (p 8, LN 358-366)
In addition, we included reliability evidence from Mao et al. (2016) and Bonaiuto et al. (2016) and elaborated on the item-screening process based on factor loadings. We trust that these revisions address the reviewer’s concerns regarding the clarity and rigor of the scale adaptation procedure.
- The authors should justify the use of different response scales, as the FKCAS is a 5-point Likert scale while the other scales in this study use a 7-point scale. How will the potential bias introduced by this discrepancy be addressed?
Response to Comments: We sincerely thank the reviewer for raising this important methodological point. Following the suggestion, we have added a clarification in the Method section and an acknowledgement in the Limitations section. In addition, we conducted an extra robustness analysis to determine whether the use of different response formats (5-point vs. 7-point) may have biased the findings.
“To address the potential influence of using different response scales across measures (the FKCAS was rated on a 5-point scale, whereas the other instruments used 7-point scales), we conducted an additional robustness check. Specifically, we re-estimated the mediation model using all items as observed indicators. The structural paths remained highly consistent with those from the latent-variable model. Key paths such as Flow → FLCA (β = –.231, p < .001), Flow → academic efficacy (β = –.373, p < .001), FLCA → academic efficacy (β = .466, p < .001), Flow → PsyCap (β = .352, p < .001), and academic efficacy → PsyCap (β = –.097, p < .001) all remained statistically significant. This convergence suggests that the discrepancy in response formats did not materially bias the substantive conclusions.”(This text has been inserted into 4.3. Test of our measurement model; p 10, LN 434-444)
“Finally, although a robustness test showed that differences in response formats (5-point vs. 7-point) did not substantially alter the direction or magnitude of the structural paths, minor scale-related inconsistencies may still exist in the current study. Future studies may benefit from using unified response formats or applying scale-harmonization techniques to minimize potential measurement bias further.”(This text has been inserted into the 5.4. Limitations and future research; p 14, LN 584-588)
While the item-level model had less optimal global fit (RMSEA = .062, CFI = .818), the structural relations were stable and aligned with the original model. We believe this additional analysis meaningfully strengthens the robustness and transparency of our results.
Results:
- A short justification for their choice of statistical analyses is needed. Why was CFA used? Why was SEM used for mediation analysis?
Response to Comments:
Thank you for your insightful comment. Confirmatory factor analysis (CFA) was used to examine the measurement model because all latent constructs (flow, academic efficacy, FLCA, and PsyCap) were measured by multiple observed indicators. CFA allowed us to assess the reliability and validity of these constructs before testing the structural relationships (Hair et al., 2010).
Structural equation modeling (SEM) was then employed to test the hypothesized relationships and mediation effects simultaneously. SEM is appropriate for this study because it enables the estimation of complex models involving multiple mediators and latent variables while accounting for measurement error (Kline, 2023). This approach provides a more rigorous and comprehensive analysis than separate regression-based methods.
We have added a brief justification of these analytic choices to the Method section of the revised manuscript for greater clarity.
“CFA allowed us to assess the reliability and validity of these constructs before testing the structural relationships (Hair et al., 2010).” (p 9 , LN 401-403)
“Finally, a structural equation modeling using AMOS (version 25) was employed to conduct a chained mediation effect test, as it allows for the simultaneous estimation of multiple latent variables and indirect effects while controlling for measurement error (Kline, 2023).” (p 9, LN 404-407)
- The authors state that “All the standardized factor loadings for the indicators of constructs were above the cut-off value (0.40) and significant (p < .001).” This is unclear. Please report the specific factor loadings for each indicator.
Response to Comments:
Thank you for your thoughtful comment. The standardized factor loadings for all indicators are already presented in Figure 2 (Measurement Model). To improve clarity, we have now explicitly mentioned this in the text.
“Specifically, the loadings ranged from .79 to .85 for flow, from .65 to .89 for FLCA, from .83 to .95 for academic efficacy, and from .77 to .96 for PsyCap (see Figure 2 for detailed values).” (p 10, LN 432-434)
- Although the fit index was reported, the authors need to indicate whether these values demonstrate an acceptable model fit based on established guidelines.
Response to Comments:
Thank you for this helpful comment. We agree that it is important to interpret the model fit indices according to established guidelines. In the revised manuscript, we have added a statement clarifying that the obtained fit indices indicate an acceptable model fit.
“The CFA results were all within the recommended thresholds (Sathyanarayana & Mohanasundaram, 2024; Wheaton et al., 1997), indicating an acceptable fit to our data: χ²/df = 4.550; GFI = 0.964; CFI = 0.982; NFI = 0.978; TLI = 0.976; SRMR = 0.036; RMSEA = 0.047.” (p 9, LN 427-430)
- The authors need to describe the path coefficients (e.g., whether the reported path coefficients are standardized or unstandardized).
Response to Comments:
Thank you for your helpful comment. We agree that it is important to specify whether the reported path coefficients are standardized or unstandardized. In the revised manuscript, we have clarified that all reported path coefficients (β) are standardized estimates derived from the SEM analysis.
“The path diagram and standardized path coefficients (β) are shown in Figure 3.” (p 11, LN 463-464)
- While specific indirect effects are given in Table 2, the total indirect effect is not explicitly stated.
Response to Comments:
Thank you for this helpful comment. In fact, Table 2 in our original manuscript already illustrates the total indirect effect. To enhance clarity, we have included a description of the proportion of the total indirect effect within the revised manuscript.
“The total indirect effect was 0.081, which is significant and accounts for approximately 11% of the total effect.” (p 11, LN 459-460)
Discussion:
- The authors restate the findings without providing a deep discussion. For example, they mentioned “flow can significantly and positively predict their PsyCap” but doesn't delve into the reasons why this might be the case, beyond a general reference to the broaden-and-build theory. While the broaden-and-build theory is mentioned, the first paragraph of the discussion does not explicitly explain how the study’s statistical findings confirm this theory.
Response to Comments:
Thank you for this valuable comment, and a deeper theoretical interpretation is needed to connect the empirical findings with the broaden-and-build theory (BBT). In the revised manuscript, we have substantially expanded the Discussion section.
“The results showed that EFL learners’ flow significantly and positively predicted their PsyCap, further supporting the BBT's view that positive emotions broaden individuals’ momentary thought and action repertoires and build enduring psychological resources (Fredrickson, 2001). Flow exemplifies a positive state in which learners’ flow helps regulate emotions and enhance engagement, and, over time, these positive emotional and cognitive expansions contribute to building more stable psychological resources, reflected in the higher levels of PsyCap observed in this study.” (p 11, LN 471-477)
- The discussion section cited a lot of studies without interpret the cited studies together with their main results. I feel like reading the literature review again.
Response to Comments:
Thank you for this constructive feedback. In the revised manuscript, we have reorganized this section to better synthesize prior research with our empirical results. Rather than listing studies, we now interpret how the current findings converge with or extend previous evidence. These revisions make the Discussion more analytical and less descriptive.
“This result is consistent with previous studies, as flow, as an optimal experience, can elicit positive emotions and enhance subjective well-being (Bassi et al., 2014; Rogatko, 2009). These studies collectively suggest that flow fosters psychological growth by generating pleasurable emotions. In line with this, the present study shows that when EFL learners experience flow in a language task, they can experience intrinsic satisfaction and academic achievement in the learning process. These positive emotional experiences help enhance their self-efficacy and sense of hope, which, in turn, increases the overall level of PsyCap.” (p 11, LN 478-481; p 12, LN 482-485)
- Although the author mentioned the limitation in their conclusion part. The discussion generally presents a positive view of the results without acknowledging any limitations or alternative explanations. A strong discussion section should critically evaluate the study's strengths and weaknesses and consider potential confounding factors. So, remove the limitation from the conclusion section and add a paragraph of limitations in the discussion.
Response to Comments:
Thank you for this valuable suggestion. We have revised the structure of the Discussion and Conclusion sections accordingly.
“5.4. Limitations and future research
Although the present study provides empirical evidence for the relationships among flow, FLCA, academic efficacy, and PsyCap in EFL learning, several limitations should be acknowledged.
Firstly, although all participants were university EFL learners taking compulsory English courses, individual learner characteristics (e.g., differences in learning engagement or academic self-beliefs) may still influence the observed relationships, which limits the generalizability of the findings. Future studies could consider more diverse learner profiles to capture these individual differences better.
Secondly, the sample selected in this study focuses on college learners, which does not adequately represent diverse cultural backgrounds or age stages, and subsequent studies can focus on other groups (e.g., secondary school students and adult education groups). Future studies may consider secondary school students, adult learners, or cross-cultural samples.
Thirdly, this study adopted a large sample cross-sectional approach. Although the results could test the association between variables, they could not show the causal direction of flow and PsyCap, which could be further verified by longitudinal tracking or experimental intervention. Future studies might use longitudinal or intervention designs to confirm the causal relationships posited in this model.
Finally, although a robustness test showed that differences in response formats (5-point vs. 7-point) did not substantially alter the direction or magnitude of the structural paths, minor scale-related inconsistencies may still exist in the current study. Future studies may benefit from using unified response formats or applying scale-harmonization techniques to minimize potential measurement bias further.” (p 13, LN 565-583; p 14, LN 584-588)
Comments on the Quality of English Language
- Kindly have the paper polished by a native English speaker who is good at academic writing to improve the clarity and the tense inconsistencies.
Response to Comments: We sincerely appreciate the reviewer’s valuable suggestion regarding language refinement. The entire manuscript has been thoroughly edited by a professional English-language editor who is a native speaker with expertise in academic writing. Particular attention was given to improving grammatical accuracy, tense consistency, and overall clarity of expression. We believe that these revisions have substantially enhanced the readability and presentation quality of the manuscript.
Reviewer 3 Report
Comments and Suggestions for Authors
The paper explores the impact of flow on EFL learners’ psychological capital (PsyCap), the mediating roles of foreign language classroom anxiety (FLCA) and academic efficacy. The introduction and theoretical part is well informing about all the relevant information about the flow, the psychological capital, FLCA and academic efficacy and burnout, as well as how they form the research aim, questions and three hypothesis. The only objection so far is the concentration on authors from eastern countries, while the paper cites mainly older work of western countries experts in the area of psycholinguistics. If authors believe Chinese participants may react differently to the questionnaire than their western countries counterparts, and that might be the reason why they concentrate mainly on the mentioned type of scientific papers and their authors, then such a hypothesis should be clearly stated. The instruments used are well descibed, as are the methods of analysing the gathered data and the participants. Thze discussion section is well organised, showing all the relevant results pointing to the conclusion that flow significantly and positively predicts the academic efficacy of EFL learners, which in turn impacts their PsyCap. The authors briefly mention the limitations of the study but it would be advisable to concentrate also on the applicability of the research findings and feasible implications.
Author Response
Reviewer #3
The paper explores the impact of flow on EFL learners’ psychological capital (PsyCap), the mediating roles of foreign language classroom anxiety (FLCA) and academic efficacy. The introduction and theoretical part is well informing about all the relevant information about the flow, the psychological capital, FLCA and academic efficacy and burnout, as well as how they form the research aim, questions and three hypothesis. The only objection so far is the concentration on authors from eastern countries, while the paper cites mainly older work of western countries experts in the area of psycholinguistics. If authors believe Chinese participants may react differently to the questionnaire than their western countries counterparts, and that might be the reason why they concentrate mainly on the mentioned type of scientific papers and their authors, then such a hypothesis should be clearly stated. The instruments used are well descibed, as are the methods of analysing the gathered data and the participants. Thze discussion section is well organised, showing all the relevant results pointing to the conclusion that flow significantly and positively predicts the academic efficacy of EFL learners, which in turn impacts their PsyCap. The authors briefly mention the limitations of the study but it would be advisable to concentrate also on the applicability of the research findings and feasible implications.
Response to Comments: We are grateful to the reviewer’s comment on the study. We did cite previous studies of western countries experts in the area of positive psychology, but one primary purpose of the study was to testify whether such scales or studies would be applicable in the Chinese context. Meanwhile, we have added this as one of the limitations, admitting that individual differences may affect the results of the current study. In addition, concerning your comments on the applicability of the research findings and feasible implications, we have added one paragraph at the last part of the conclusion:
“In a practical sense, this study provides important insights for optimizing EFL teaching strategies. First, to foster students’ flow—as a critical pathway to developing PsyCap—EFL teachers can assess proficiency level by assigning tasks (e.g., sentence filling, vocabulary matching), and then tailor task difficulty to achieve a balance between challenge and skill. This balance not only sustains engagement, but also strengthens learners’ sense of efficacy and optimism. Second, design flow-conducive environments—such as integrating gamified learning, cooperative goal-setting, and peer-supported activities—can mitigate FLCA. A balanced supportive-challenging environment encourages students to take academic risks, thereby facilitating the internalization of academic efficacy and the growth of resilience. These strategies collectively contribute to a virtuous cycle where positive emotions, reduced anxiety and burnout, and enhanced self-beliefs support sustainable language learning outcomes.”
(P14, Line 586-597)